# Stabilization of a molecular water oxidation catalyst on a dye—sensitized photoanode by a pyridyl anchor

Yong Zhu[1], Degao Wang[2,3], Qing Huang [2], Jian Du[1], Licheng Sun [1,4,5], Fei Li [1✉] & Thomas J. Meyer [3✉]

Understanding and controlling the properties of water-splitting assemblies in dye-sensitized photoelectrosynthesis cells is a key to the exploitation of their properties. We demonstrate here that, following surface loading of a $[Ru(bpy)_3]^{2+}$ (bpy = 2,2'-bipyridine) chromophore on nanoparticle electrodes, addition of the molecular catalysts, $Ru(bda)(L)_2$ (bda = 2,2'-bipyridine-6,6'-dicarboxylate) with phosphonate or pyridyl sites for water oxidation, gives surfaces with a 5:1 chromophore to catalyst ratio. Addition of the surface-bound phosphonate derivatives with L = 4-pyridyl phosphonic acid or diethyl 3-(pyridin-4-yloxy)decyl-phosphonic acid, leads to well-defined surfaces but, following oxidation to Ru(III), they undergo facile, on-surface dimerization to give surface-bound, oxo-bridged dimers. The dimers have a diminished reactivity toward water oxidation compared to related monomers in solution. By contrast, immobilization of the Ru-bda catalyst on $TiO_2$ with the 4,4'-dipyridyl anchoring ligand can maintain the monomeric structure of catalyst and gives relatively stable photoanodes with photocurrents that reach to 1.7 mA cm$^{-2}$ with an optimized, applied bias photon-to-current efficiency of 1.5%.

[1] State Key Laboratory of Fine Chemicals, Dalian University of Technology, Dalian 116024, China. [2] Engineering Laboratory of Advanced Energy Materials, Ningbo Institute of Industrial Technology, Chinese Academy of Sciences, Ningbo, Zhejiang 315201, China. [3] Department of Chemistry, University of North Carolina at Chapel Hill, Chapel Hill, NC 27599, USA. [4] Department of Chemistry, School of Engineering Sciences in Chemistry, Biotechnology and Health, KTH Royal Institute of Technology, Stockholm 10044, Sweden. [5] Center of Artificial Photosynthesis for Solar Fuels, School of Science, Westlake University, Hangzhou 310024, China. ✉email: lifei@dlut.edu.cn; tjmeyer@unc.edu

In solar powered water splitting, water oxidation is a mechanistic bottleneck[1]. The emergence of molecular water oxidation catalysts (WOCs) has provided an opportunity to mimic the oxygen-evolving complex (OEC) of photosystem II in nature[2,3]. In one approach, the dye-sensitized photoelectrosynthesis cell (DSPEC)[4–6], a molecular chromophore and catalyst are immobilized on wide bandgap, n-type semiconductors, such as fluorine-doped tin oxide (FTO), with a mesoporous $TiO_2$ layer. Water oxidation is trigged by sequential absorption of four photons with the accumulation of oxidative equivalents at a catalyst. Complexes of the type, $Ru(bda)(L)_2$ (bda = 2,2′-bipyridine-6,6′-dicarboxylate) with L a pyridyl ligand, have been shown to be efficient catalysts for homogenous or heterogeneous water oxidation with both favorable kinetics and low overpotentials for water oxidation[7,8]. Although the catalytic mechanism for Ru-bda water oxidation has been well studied in solution, there is less insight on electrode surfaces.

A notable difference between homogeneous and heterogenized catalysts is the surface anchoring group. Apart from providing a basis for surface immobilization, the bridging ligand can have a significant impact on electron and hole transfer through a metal oxide/molecular interface[9,10]. In a DSPEC, electronic coupling through a bridging ligand enables efficient electron injection from a photo-excited dye into the conduction band of the semiconductor. By contrast, a surface catalyst for water oxidation, requires long-lived charge separation and the accumulation of four oxidative equivalents with a different distance behavior between activation and back electron transfer. Although a variety of anchors have been explored in dye-sensitized solar cells (DSSCs)[11], suitable candidates for DSPEC photoanodes are dictated by the requirements for electron transfer and hydrolytic stability. To date, phosphonic and carboxylic acids have been the most widely used anchoring groups, especially phosphonates because of their hydrolytic stability on $TiO_2$[4,12–19]. The acid anchors prefer to attach on $TiO_2$ by covalent bonding with the hydroxyl groups (Brønsted acid sites) on the surface. Though stable in organic media, the stability of covalent binding in aqueous solution is largely dependent on external pH values because of potential hydrolysis at pH > 5. This undesirable hydrolysis results in the detachment of chromophore or catalyst from the metal oxide surface.

In comparison, pyridine is known to adsorb over $TiO_2$ surface by coordination with the exposed Ti atoms (Lewis acid sites). Given the different mechanisms between phosphonate and pyridine binding, pyridine anchored $TiO_2$ was expected to be more stable against hydrolysis in near-neutral aqueous solutions. In the literature, catalysis with pyridine as an anchor for surface-bound catalysts is scarcely reported. Sakai and Ozawa et al.[20] have constructed a photoanode with a pyridine-functionalized [Ru(bpy)₃]²⁺ (bpy = 2,2′-bipyridine) complex as the chromophore immobilized on $TiO_2$ film. This photoanode was examined for the PEC oxidation of EDTA. Comparison with analogous phosphonate-derivatized [Ru(bpy)₃]²⁺ complexes demonstrates

improved surface stability in aqueous solution at pH 5. Another example was also reported by the same group, where a pyridine-derivatized platinum(II) porphyrin was used as an electrocatalyst for water reduction[21].

In this study, we demonstrate, an important dependence for a surface-bound catalyst for water oxidation on surface anchoring groups. The results described here show that upon oxidation, a Ru-bda catalyst with phosphonate binding, can rapidly form surface-bound μ-oxo-bridged dimers which have a greatly decreased reactivity toward water oxidation. We also find that the dimerization process can effectively be inhibited by replacing phosphonate acid binding groups with a pyridine anchor. The impact of the structural change on the performance of dye-sensitized photoanodes is remarkable. By utilizing pyridine binding, the photocurrent is enhanced by an order of magnitude giving a maximum solar photon-to-current efficiency for water oxidation of 1.5%.

## Results

**Preparation of electrodes.** In the research described here, photoanodes were prepared by co-adsorption of a phosphonate-derivatized tris(2,2′-bipyridine)ruthenium(II) dichloride salt of (**RuP**²⁺) as the chromophore with pyridyl-derivatized Ru(bda)(4,4′-bpy)₂ (4,4′-bpy = 4,4′-bipyridine, **1**) catalysts on the surfaces of a post-treated $TiO_2$ layer on FTO (Fig. 1a). Complex **1** was synthesized in a one-step reaction between $Ru(bda)(DMSO)_2$ the 4,4′-bpy in a procedure that is far more facile than for phosphonate[22] or silane linking (Supporting Information)[17,23]. The electrodes consisted of 6 μm layers of 20 nm nanoparticles of $TiO_2$ on FTO substrates (Supplementary Fig. 5). Chromophore and catalyst were loaded on $TiO_2$ by soaking the slide in methanol solutions of **RuP**²⁺ (1 mM) and catalyst (1 mM) in sequence. As noted below, spectrophotometric analysis of the electrodes reveals that, under the conditions used for surface loading, the ratio of chromophore to catalyst was ~5:1 with the external surface dominated by the chromophore[24,25]. As a way to demonstrate and modify the surface anchoring group, the phosphonate-bound catalysts, $Ru(bda)(4-pyPO_3H)_2$ (4-pyPO₃H is 4-pyridyl phosphonic acid) (**2**), and $Ru(bda)(4-pyO(CH_2CH_2)_5PO_3H)_2$ (4-pyO(CH₂CH₂)₅PO₃H is diethyl 3-(pyridin-4-yloxy)decyl-phosphonic acid) (**3**)[16] were also prepared (Fig. 1b).

The electrochemical properties of co-derivatized films of the catalysts and **RuP**²⁺ on $TiO_2$ were investigated by cyclic voltammetry. In a pH 5.8 acetate buffer solution, the electrode $TiO_2$|-(**RuP**²⁺)₅,**1** includes a $2e^-$, $Ru^{IV}\text{-}OH^+/Ru^{II}\text{-}OH_2^{2+}$ couple at $E_{1/2} \sim 0.7$ V vs. NHE (Fig. 2)[26,27]. A catalytic current for water oxidation appeared at 1.1–1.2 V with a significant current enhancement appearing at a higher potential due to the onset wave for the **RuP**³⁺/²⁺ couple at $E_{1/2} \sim 1.25$ V.

Based on an analysis of peak currents and inductively coupled plasma emission measurements[14,28], surface loadings of

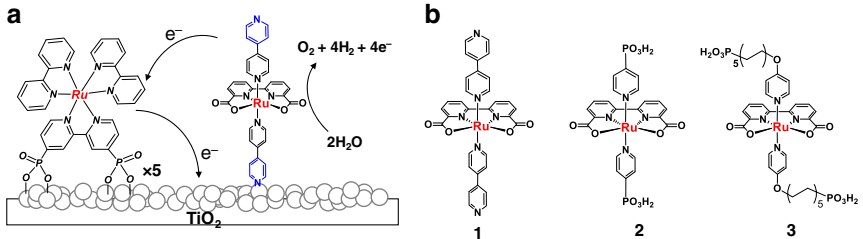

**Fig. 1 Photoanode schematic and catalysts structure. a** A surface loading scheme for cross-surface electron transfer between **RuP**²⁺ and catalyst **1** co-loaded on a photoanode. **b** Structures of the catalysts used in the study.

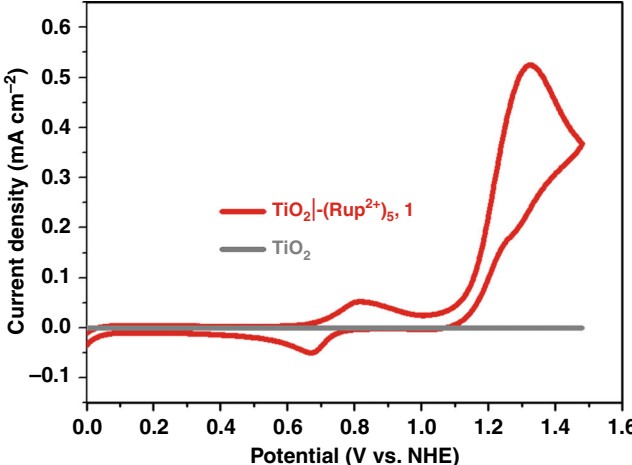

**Fig. 2 Cyclic voltammetry.** Cyclic voltammogram for TiO$_2$ and TiO$_2$|-($\mathbf{RuP}^{2+}$)$_5$,**1** in 0.1 M acetic acid/acetate buffer at pH 5.8 in 0.5 M NaClO$_4$ at a scan rate 10 mV s$^{-1}$.

chromophore and catalyst on the photoanode were $1.2 \times 10^{-8}$ mol cm$^{-2}$ for **1** and $6 \times 10^{-8}$ mol cm$^{-2}$ for $\mathbf{RuP}^{2+}$, respectively, consistent with a chromophore to catalyst ratio of 5:1 (Supplementary Fig. 6a, b). A related CV behavior was observed for the electrode TiO$_2$|-($\mathbf{RuP}^{2+}$)$_5$,**2** with a surface loading of $6 \times 10^{-8}$ mol cm$^{-2}$ for $\mathbf{RuP}^{2+}$ and $1.3 \times 10^{-8}$ mol cm$^{-2}$ for **2** (Supplementary Fig. 6c), which are comparable to that for TiO$_2$|-($\mathbf{RuP}^{2+}$)$_5$,**1**. The corresponding redox potential for the Ru$^{IV}$-OH$^+$/Ru$^{II}$-OH$_2$$^{2+}$ couple in TiO$_2$|-($\mathbf{RuP}^{2+}$)$_5$,**2** was $E_{1/2} \sim 0.65$ V with a catalytic onset at 1.1–1.2 V (Supplementary Fig. 7). The surface loading for TiO$_2$|-($\mathbf{RuP}^{2+}$)$_5$,**3** was $1.2 \times 10^{-8}$ mol cm$^{-2}$ for **3** (Supplementary Fig. 6d) with $E_{1/2} \sim 0.65$ V (Supplementary Fig. 8). However, as noted below, for both of the phosphonate catalysts, CV scans to the higher oxidation states result in dimerization of the catalyst with a significant decrease in catalytic ability.

**Water splitting.** Photoelectrochemical measurements and water splitting were investigated for TiO$_2$|-($\mathbf{RuP}^{2+}$)$_5$,**1** in a pH 5.8 acetate buffer in 0.5 M NaClO$_4$. In these experiments the cathode was a Pt mesh counter electrode with Ag/AgCl as the reference electrode. The light source was simulated sunlight (300 W Xenon arc lamp at a light intensity of 100 mA cm$^{-2}$ equipped with a 400-nm cut-off filter)[4]. As shown by photoelectrochemical linear scan voltammograms (LSV), in Fig. 3a, visible light irradiation of TiO$_2$|-($\mathbf{RuP}^{2+}$)$_5$,**1** resulted in a rapid increase in photocurrent at an onset potential of −0.2 V vs. NHE which reached a maximum value of 1.7 mA cm$^{-2}$ at 0.1 V vs. NHE. Consistent results were obtained over extended periods by using chopped light irradiation at 30 s intervals with a constant bias (Fig. 3b). Conversely, for the assembly TiO$_2$|-($\mathbf{RuP}^{2+}$)$_5$,**2**, the maximum current was reduced by a factor of 12 to 140 μA cm$^{-2}$, Fig. 3a.

As shown in Table 1, the photocurrent for TiO$_2$|-($\mathbf{RuP}^{2+}$)$_5$,**1** exceeds values for other dye-sensitized photoanodes under similar conditions. An incident photo-to-current efficiency (IPCE) of 25% was obtained for TiO$_2$|-($\mathbf{RuP}^{2+}$)$_5$,**1**, at the visible absorption maximum for $\mathbf{RuP}^{2+}$ (Fig. 3c), a value that is comparable to the highest IPCE values that have been documented for photoanodes of this kind (Table 1)[16]. Compared to other catalyst design structures, the results here are notable because of the ease of electrode preparation and cell performance[29,30]. Preparation of the electrode involves a simple

electrode fabrication step with a doctor-blade-coated electrode followed by addition of the catalyst[17,31].

After a 2-h irradiation period for TiO$_2$|-($\mathbf{RuP}^{2+}$)$_5$,**1** at 0.2 V vs. NHE, the photocurrent had fallen to 0.8 mA cm$^{-2}$, which is also impressive compared to related literature examples (Supplementary Fig. 9). During this process, 6.24 μmol evolved oxygen was quantified by gas chromatography with 2.66C charges passed through the photoanode, corresponding to a faradaic efficiency of over 90% and a TON of 520. Hydrogen at the cathode was also monitored by gas chromatography and shown to be produced in the 2:1 ratio consistent with water oxidation (Supplementary Fig. 10).

An additional parameter of note in characterizing the electrodes is the applied bias photon-to-current efficiency (ABPE) which has not been systematically investigated for electrodes of this type[32]. As shown in Fig. 3d, the optimal ABPE for TiO$_2$|-($\mathbf{RuP}^{2+}$)$_5$,**1** was 1.5% at 0.3 V vs. NHE based the photocurrents from a 1 sun, AM 1.5 G light source. The values reported here are lower than values reported for the visible-light-absorbing semiconductor photoanodes BiVO$_4$ and Ta$_3$N$_5$ (2.5% at 0.56 V for BiVO$_4$[33] and 2.5% at 0.9 V for Ta$_3$N$_5$[34]). However, they appear at relatively low applied potentials because of the more negative conduction band potential of TiO$_2$ compared to the narrow bandgap semiconductors which is of advantage in bias-free water splitting with an external photocathode[35,36].

**Inhibition of back electron transfer.** The results from the PEC water oxidation experiments point to an important role for the bridging ligand. To further understand the effect of the anchoring ligand, photoelectrochemical impedance spectroscopy measurements (PEIS) were conducted. Nyquist plots, and an appropriate equivalent circuit diagram, are shown in Fig. 4a. A typical PEIS spectrum for DSPEC exhibits a semicircle in the Nyquist plots, arising from charge recombination resistance ($R_{rec}$) at the interface between the TiO$_2$|chromophore and the catalyst[31,37]. Given the comparable loadings of the chromophores on the electrodes, the semicircles in the PEIS measurements directly reflect the capability of the catalyst to inhibit back electron transfer. As shown in Fig. 4a, the Nyquist plot for TiO$_2$|-$\mathbf{RuP}^{2+}$,**1** (348 Ω) includes a larger semicircle radius compared for TiO$_2$|-$\mathbf{RuP}^{2+}$,**2** (155 Ω). The electron recombination rate constant, $k_r$, can be estimated from, $k_r = 1/C \cdot R_{rec}$, with the corresponding $K_r$ values for photoanodes from catalyst **1** and **2** of 56 and 146 s$^{-1}$ consistent with a role for the pyridyl bridging ligand in reducing back electron transfer.

Given these results, a Ru-bda derived catalyst, decorated with decyl-phosphonic acid, Ru(bda)(4-pyO(CH$_2$CH$_2$)$_5$PO$_3$H)$_2$ (**3**), was also co-loaded with $\mathbf{RuP}^{2+}$ on TiO$_2$ (Fig. 1)[16]. Due to the long alkyl spacer, the catalyst center in the assembly is predictably well-separated from the oxide surface, minimizing back electron transfer[38,39]. As expected, a large semicircle (1453 Ω) in the high frequency region of PEIS Nyquist plot was observed for TiO$_2$|-$\mathbf{RuP}^{2+}$,**3**, consistent with a much smaller recombination rate, 35 s$^{-1}$, compared to 56 s$^{-1}$ for **1** and 146 s$^{-1}$ for **2**. Even with this kinetic advantage, the PEC experiment showed that TiO$_2$|-($\mathbf{RuP}^{2+}$)$_5$,**3** produced a photocurrent at 0.8 mA cm$^{-2}$, only one-half that for TiO$_2$|-$\mathbf{RuP}^{2+}$,**1** under the same conditions (Fig. 4b).

**Dimerization.** During the PEC experiments, TiO$_2$|-($\mathbf{RuP}^{2+}$)$_5$,**2** (Fig. 5a) and TiO$_2$|-($\mathbf{RuP}^{2+}$)$_5$,**3** (Supplementary Fig. 11) underwent a rapid color change from dark red to green with the appearance of strong absorption bands at 700 nm. Appearance of the new absorption feature, as noted above, was consistent with the results of previous studies and the appearance of surface-

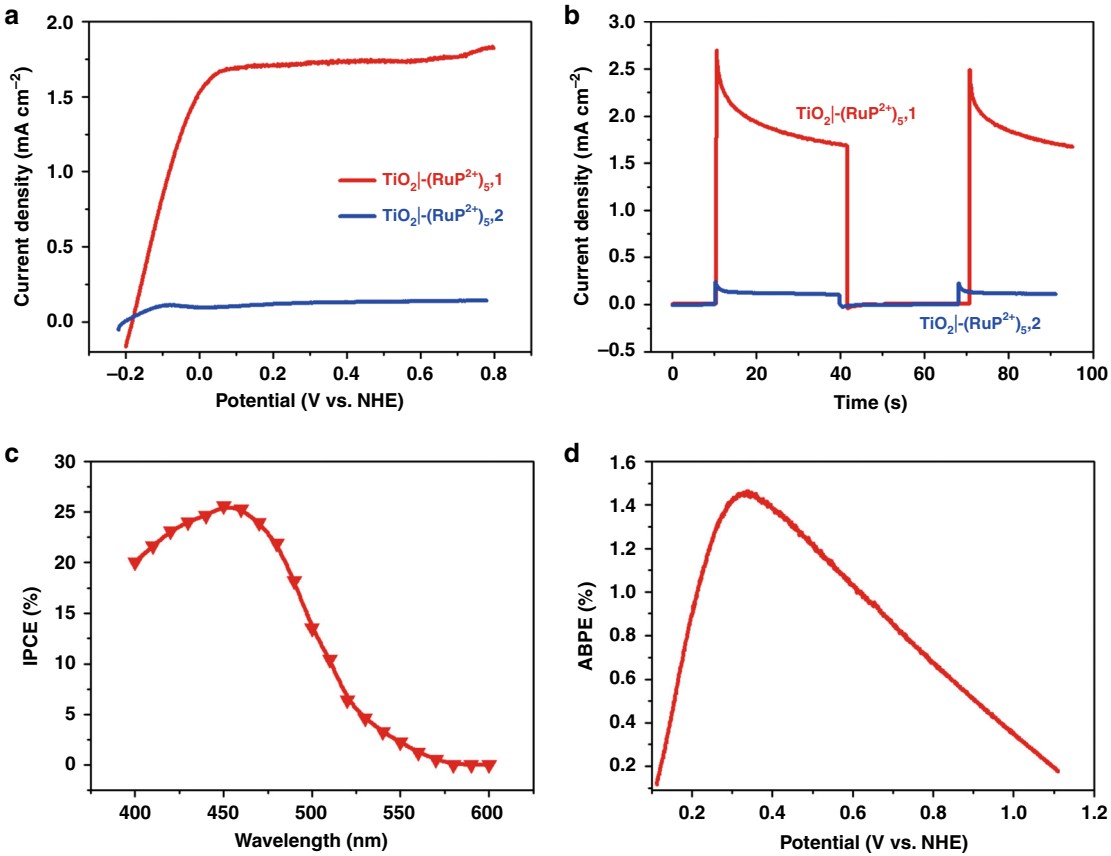

**Fig. 3 Photocurrent and efficiencies of photoanodes. a** Potential dependence of photocurrents for $TiO_2|\text{-}(\textbf{RuP}^{2+})_5,\textbf{1}$ and $TiO_2|\text{-}(\textbf{RuP}^{2+})_5,\textbf{2}$ with illumination above 400 nm at 100 mW cm$^{-2}$ at pH 5.8 in a 0.1-M acetate buffer containing 0.5 M NaClO$_4$. **b** As in **a**, current density-time traces for $TiO_2|\text{-}$ $(\textbf{RuP}^{2+})_5,\textbf{1}$ and $TiO_2|\text{-}(\textbf{RuP}^{2+})_5,\textbf{2}$ at a constant bias of 0.2 V vs. NHE. **c** Incident photon-to-current efficiencies (IPCEs) for $TiO_2|\text{-}(\textbf{RuP}^{2+})_5,\textbf{1}$ at 0.2 V vs. NHE. **d** Applied bias photon-to-current efficiencies (ABPEs) for $TiO_2|\text{-}(\textbf{RuP}^{2+})_5,\textbf{1}$ for photoelectrochemical water splitting.

**Table 1 Relative efficiencies for water oxidation by DSPEC photoanodes[a].**

| Description of photoanode | pH | Maximum IPCE (%) | Maximum stable photocurrent density (mA/cm$^2$) | Substrate | Ref. |
|---|---|---|---|---|---|
| $RuP^{2+}$-$Zr^{4+}$-Ru-bda assembly | 5.7 | – | 1–1.5 | SnO$_2$/TiO$_2$ core-shell | 43 |
| $RuP^{2+}$/Ru-bda co-loaded | 5.7 | – | 0.97–1.45 | | |
| $RuP^{2+}$/Ru(bda)(L$_{O\text{-}C10}$) co-loaded[b] | 7 | 24.8 | 1.4 | SnO$_2$/TiO$_2$ core-shell | 16 |
| | 4.7 | 15.3 | 0.8 | | |
| $RuP^{2+}$-Ru-bda assembly | 5.7 | – | 0.85 | SnO$_2$/TiO$_2$ core-shell | 44 |
| $RuP^{2+}$-Ru-bda assembly | 7 | 3.1 | 0.4 | SnO$_2$/TiO$_2$ core-shell | 45 |
| $RuP^{2+}$-ALD SnO$_2$-Ru-bda assembly | 4.7 | 17.1 | 0.85 | SnO$_2$/TiO$_2$ core-shell | 4 |
| TPA/Ru-bda co-loaded | 4.8 | 0.3 | 0.4 | SnO$_2$/TiO$_2$|Al$_2$O$_3$ | 15 |
| $RuP^{2+}$/IrO$_2$ co-loaded | 5.7 | – | 0.03 | TiO$_2$ nanoparticle film | 12 |
| $RuP^{2+}$/IrO$_2$ co-loaded[c] | 5.8 | – | 0.08 | TiCl$_4$ treated TiO$_2$ | 13 |
| Zn Porphyrin/IrCp* co-loaded[d] | 7 | – | 0.03 | TiO$_2$ nanoparticle film | 18 |
| $RuP^{2+}$/Ru-bda co-loaded | 5.8 | 25 | 1.7 | TiCl$_4$ treated TiO$_2$ | This work |

[a]Unless otherwise specified, simulated sunlight with a density of 100 mW cm$^{-2}$ was used as the light source.
[b]L$_{O\text{-}C10}$ is diethyl 3-(pyridin-4-yloxy)decyl-phosphonic acid.
[c]A 150-W Xe lamp was used as the light source with a 410 nm, long-pass filter and a water filter.
[d]At 200 mW cm$^{-2}$.

bound μ-oxo bridged, binuclear ruthenium dimers as shown in Fig. 6 for [Ru(bda)(4-pyPO$_3$H)$_2$]$_2$O (**2′**)[40,41].

The appearance of the dimer was investigated by cyclic voltammetry on *nano*ITO films. As shown in Fig. 4c, complete conversion from **2** to **2′** occurred after four CV scan cycles. The redox features for **2′** closely resemble those reported previously for a surface-bound Ru-bda dimer with a 3-(pyridin-4-yloxy

propyl)phosphonic acid anchor[41]. In the CV there are three single-electron processes that can be assigned to the Ru$^{II}$-O-Ru$^{III}$/Ru$^{III}$-O-Ru$^{III}$ couple (0.73 V vs. NHE), the Ru$^{III}$-O-Ru$^{III}$/Ru$^{III}$-O-Ru$^{IV}$ couple (0.93 V vs. NHE), and the Ru$^{III}$-O-Ru$^{IV}$/Ru$^{IV}$-O-Ru$^{IV}$ couple (1.3 V vs. NHE).

Compared to the monomeric precursor **2**, the decrease in catalytic current for **2′** demonstrates a reduced reactivity towards

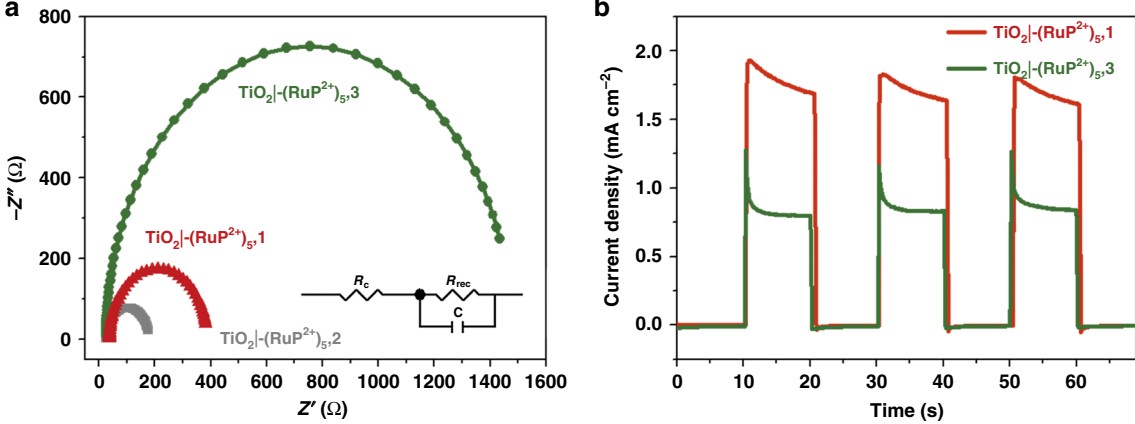

**Fig. 4 Photoelectrochemical impedance spectroscopy and photocurrent of photoanodes. a** PEIS Nyquist plots for TiO$_2$|-(**RuP**$^{2+}$)$_5$,**1**, TiO$_2$|-(**RuP**$^{2+}$)$_5$,**2**, and TiO$_2$|-(**RuP**$^{2+}$)$_5$,**3** with 420-nm LED illumination at an intensity of 3.5 mW cm$^{-2}$. Inset: the equivalent circuit used for data fitting. **b** Current density-time traces for TiO$_2$|-(**RuP**$^{2+}$)$_5$,**1** and TiO$_2$|-(**RuP**$^{2+}$)$_5$,**3** at a constant bias of 0.2 V vs. NHE under illumination (<400 nm, 100 mW cm$^{-2}$).

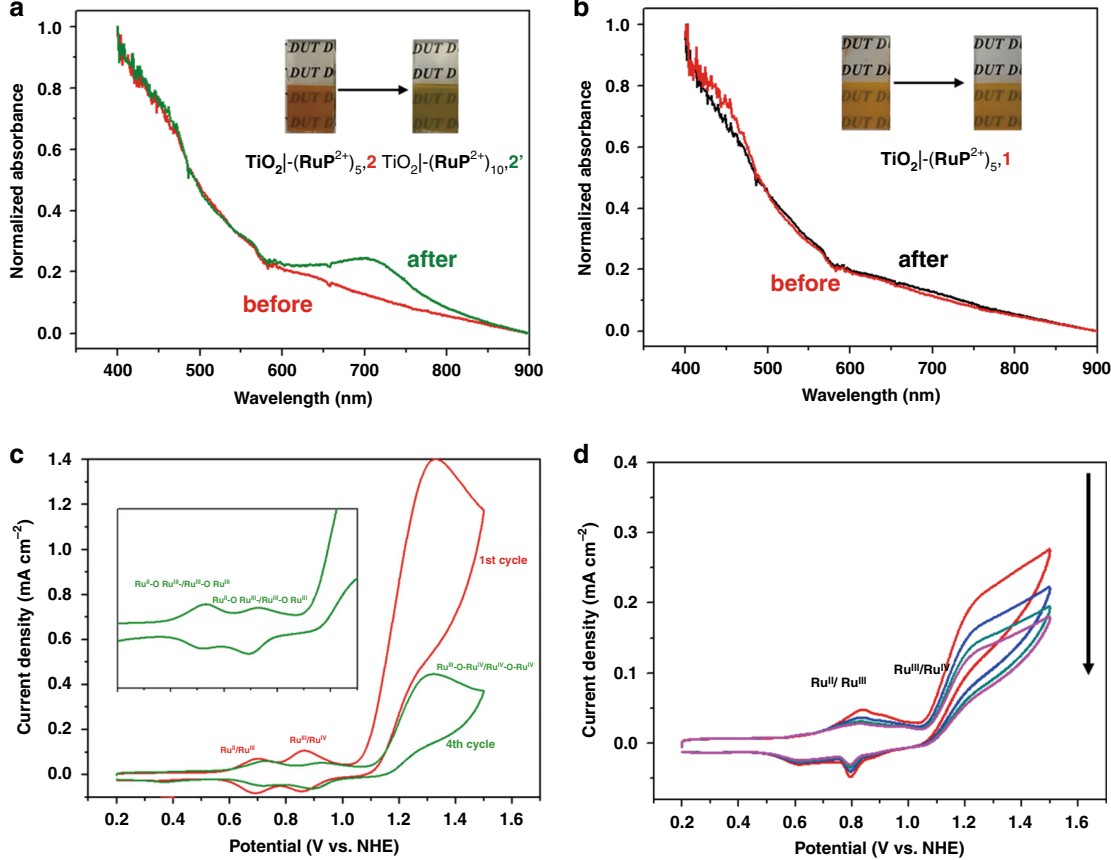

**Fig. 5 UV-Vis spectral changes of photoanodes and successive cyclic voltammogram changes.** UV-Vis spectral changes in acetate buffers (pH 5.8, 0.5 M NaClO$_4$) at a scan rate of 10 mV/s for TiO$_2$|-(**RuP**$^{2+}$)$_5$,**2** (**a**) and TiO$_2$|-(**RuP**$^{2+}$)$_5$,**1** (**b**) before and after PEC experiments. Inset: images of TiO$_2$|-(**RuP**$^{2+}$)$_5$,**2** and TiO$_2$|-(**RuP**$^{2+}$)$_5$,**1** before (left) and after the PEC experiments (right). **c** Successive cyclic voltammograms for *nano*ITO|-**2** with the insert showing results for a fourth CV cycle. (**d**) As in **c**, but for *nano*ITO|-**1**, with the decrease in current due to slow desoption of **1** from the electrode.

water oxidation with low photocurrents for TiO$_2$|-(**RuP**$^{2+}$)$_5$,**2'** (40 µA cm$^{-2}$, Supplementary Fig. 12). A similar phenomenon was also observed for TiO$_2$|-(**RuP**$^{2+}$)$_5$,**3** (Supplementary Fig. 13). In contrast to **2** and **3**, catalyst **1**, with bridging pyridyl anchoring groups resulted in no color change during the photoelectrolysis period (Fig. 5b) with stable waveforms recorded for multiple CV scans (Fig. 5d), consistent with a stable monomer structure on the electrode. The color change and dimer formation also occurred

for films of TiO$_2$|-(**RuP**$^{2+}$)$_5$,**2** and TiO$_2$|-(**RuP**$^{2+}$)$_5$,**3** in air, while TiO$_2$|-(**RuP**$^{2+}$)$_5$,**1** was relatively stable for weeks.

As noted above, the CV results are consistent with the formation of µ-oxo-bridged, Ru$^{III}$-O-Ru$^{III}$ dimers following oxidation of Ru$^{II}$ to Ru$^{III}$ in the catalyst[40,41]. Reducing the amounts of catalyst **2** or **3** that were surface loaded, was not sufficient to avoid formation of "green dimers" as shown by CV scans. In this case, the difference in surface stability between the

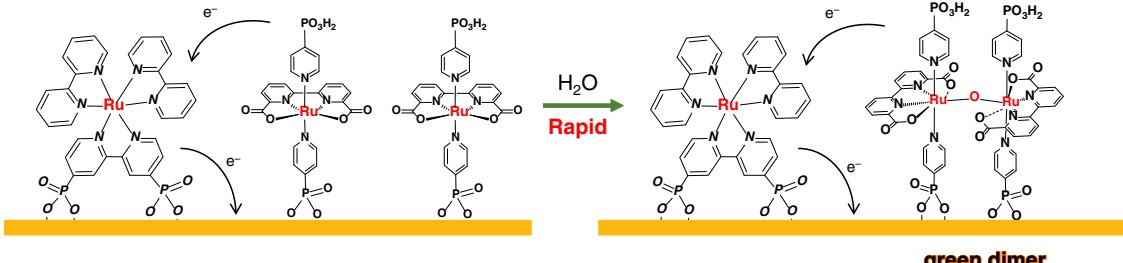

**Fig. 6 Green dimer.** Dimerization of catalyst **2** on the electrode surface.

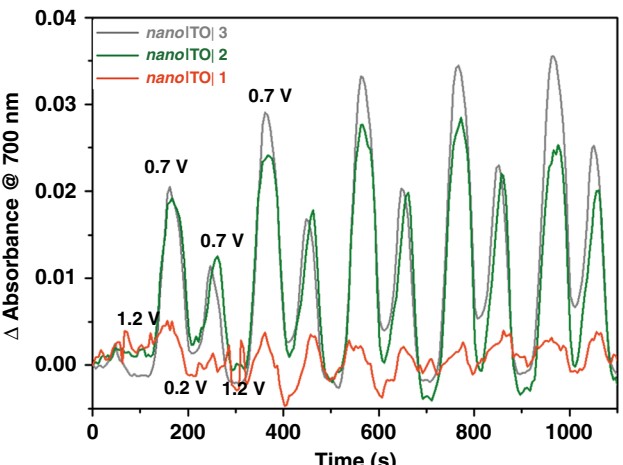

**Fig. 7 Spectroelectrochemical of photoanodes.** Absorption spectral changes at 700 nm during cyclic voltammetry scans for *nano*ITO|-**1**, **2**, and **3** at pH 5.8 in a 0.1 M acetate buffers containing 0.5 M NaClO$_4$ at a scan rate 10 mV/s.

phosphine and pyridyl ligands is significant, and presumably arises from rapid phosphonate loss from the surface in weakly acidic solutions.

The surface assemblies were also investigated spectroelectrochemical. Figure 7 shows the dynamic changes that occur at 700 nm following successive CV scans for *nano*ITO|−**1**, **2** and **3** in the dark. In these experiments, the potential was scanned in the anodic direction at 10 mV s$^{-1}$ from 0.2 to 1.2 V and then back to 0.2 V with spectra recorded every 5 s. For *nano*ITO|-**2**, after the first positive half-scan from 0.2 to 1.2 V, the absorption at 700 nm appears periodically at each half-scan, finally reaching a maximum at the peak potential for the Ru$^{II}$-O-Ru$^{III}$/Ru$^{III}$-O-Ru$^{III}$ couple. The data are consistent with the appearance of the Ru$^{III}$-O-Ru$^{III}$ form of the dimer as an active intermediate in the catalytic cycle. Similar behavior was exhibited by catalyst **3** as shown in Fig. 7. By contrast, there was no spectrophotometric evidence, for dimerization by **1**, Fig. 7, presumably because of more stable pyridine-to-surface bonding.

The dimerization of the phosphonic acid-modified Mn catalyst on mesoporous TiO$_2$ has been previously reported by Reisner et. al.[42]. The lability of immobilized **1**, **2**, and **3** were evaluated by catalyst desorption from a pH 5.8 acetate buffer containing 2% sodium ascorbate. Experimental results in Supplementary Fig. 14 reveal that catalysts **2** and **3** decorated with phosphonate anchor are prone to detach from the electrode surface with desorption rate constants $k_{des}$ of 270 s$^{-1}$ for **2** and 63 s$^{-1}$ for **3**, while catalyst **1** decorated with pyridine anchor is much more stable on the surface ($k_{des}$ (**1**) = 1.4 s$^{-1}$). Given the dynamic TiO$_2$|catalyst interface, the formation of surface-bound "green dimer" is assumed to be a result of desorption of oxidized phosphonate-

modified catalysts under the conditions of PEC experiments, followed by dimerization and re-adsorption within mesoporous TiO$_2$.

## Discussion

We have described here a straightforward procedure for the preparation of photoelectrodes for water oxidation in aqueous solutions with added acetate buffers. The surfaces of nanoparticle electrodes sensitized by a phosphonate-derivatized,Ru(II)bpy-based chromophores, were added a water oxidation catalyst based on derivatives of the Ru(bda)(L)$_2$ with the axial ligands 4,4′-bipyridine (1), 4-pyPO$_3$H (2), and 4-pyO(CH$_2$CH$_2$)$_5$PO$_3$H(3). In forming the catalytic surfaces, the chelate diphosphonate linkages of the chromophore result in complete surface coverage with subsequent addition of the mono-dentate pyridyl or phosphonate-derivatized catalysts to give final surface coverages of chromophore to catalyst of 5:1.

Cyclic voltammetry and UV-Vis spectroscopy measurements in 0.1 M acetate buffers in 0.5 M NaClO$_4$ show that oxidation of the surface-bound catalysts results in oxidation from Ru(II) to Ru(IV). Multiple scan voltammograms for the phosphonate derivatives were consistent with dimerization of the phosphonate-derivatized catalysts to O-bridged catalyst dimers. The latter were far less reactive toward water oxidation but are retained on the oxide surfaces. In a relative sense, the 4,4′-bpy electrode described here is impressive for water oxidation. The incident photo-to-current efficiency for TiO$_2$|-(**RuP$^{2+}$**)$_5$,**1**, is comparable to the highest IPCE values documented for photoanodes of this kind (Table 1) at the visible absorption maximum for **RuP$^{2+}$** (Fig. 1c), with a maximum solar efficiency of 1.5%. Our results offer an important message for molecular level assembly design. Recognizing the importance of anchoring groups in determining the catalytic active species on metal oxide surface adds a new approach to the rational tuning of hybrid interfaces for DSPEC applications.

## Methods

**Preparation of photoanode.** Ca. 6 μm thickness TiO$_2$ films were coated on FTO glass by the doctor-blade method. The TiO$_2$ paste was doctor-bladed onto a clean FTO substrate followed by a sintering process at 120 °C for 30 min, 450 °C for 1 h. After cooling to room temperature, the electrodes were immersed into a 50-mM aqueous TiCl$_4$ solution for 1 h at 70 °C before calcination at 450 °C for 1 h. The FTO supported TiO$_2$ film was sensitized by soaking the slide in a methanol solution containing **RuP** (1 mM) for 1 h, followed by soaking in another methanol solution containing complex **1** (1 mM) or complex **2** or **3** (1 mM) for 1 h. The sensitized electrode was dried in dark at room temperature after rinsing by methanol. The transparent conductive *nano*ITO thin films were prepared according to previously published methods[16].

**Photoelectrochemical measurements.** All photoelectrochemical measurements were carried out at room temperature by using a CHI 660E electrochemical workstation (Shanghai Chenhua Instrument Co., LTD). The photoelectrochemical performances of photoanodes were measured with a typical three-electrode configuration with the photoanode as the working electrode, a platinum wire as the counter electrode, and saturated Ag/AgCl (0.194 V vs. NHE) as the reference electrode. The simulated sunlight was obtained by passing light from a 300-W

Xenon arc lamp (CEAULIGHT) equipped with a 400-nm filter or an AM1.5 G filter, the power intensity of the incident light was calibrated to 100 mW/cm$^2$ using a THORLABS S401C power meter.

The incident photon to current efficiency (IPCE) at each wavelength was determined by using illumination from a 300-W Xenon arc lamp (CEAULIGHT). The monochromatic light was produced using a monochromator (Beijing 7-star optical instruments-7ISW75) with a 10-nm bandpass. The light intensity ($P_\lambda$) at each wavelength ($\lambda$) was determined with a THORLABS S120VC power meter. The IPCE value was calculated using the equation:

$$\text{IPCE}(\%) = \frac{1240 \times (J_{\text{light}} - J_{\text{dark}})}{\lambda \times P_\lambda} \times 100\% \qquad (1)$$

The applied bias photo-to-current efficiency (ABPE) was calculated based on the current-potential curves under 1 sun, AM1.5 G illumination,

$$\text{ABPE}(\%) = \frac{(J_{\text{light}} - J_{\text{dark}}) \times (1.23 - V_{\text{RHE}})}{P_{\text{light}}} \times 100\% \qquad (2)$$

With $V_{\text{RHE}}$ is the applied potential versus RHE, $J_{\text{light}}$ and $J_{\text{dark}}$ the measured photocurrent and dark current, respectively. $P_{\text{light}}$ is the power density of the lamp, AM 1.5 G (100 mW cm$^{-2}$).

**Determination of Faradaic efficiency.** The amounts of oxygen and hydrogen evolution were determined by gas chromatography. Before the measurement, the sealed electrolytic cell was degassed with Argon for 15 min to remove residual air. The amount of oxygen and hydrogen evolution after 2 h of electrolysis at 0.2 V vs. NHE was measured. The Faradaic efficiency was calculated according to the integrated charge (Q) passed and the amount of O$_2$ evolved using the equation:

$$\eta(\%) = \frac{96485 \times n_{\text{O}_2} \times 4}{Q} \times 100\% \qquad (3)$$

**Photoelectrochemical impedance spectroscopy.** PEIS was measured with typical three-electrode under an intensity of 3.5 mW cm$^{-2}$ 420-nm LED illumination at pH 5.8 in 0.1 M acetic acid/acetate buffer containing 0.5 M NaClO$_4$. The PEIS measurements were scanned from $1 \times 10^3$ to 1 Hz. PEIS experiments were carried out in galvanostatic mode at open-circuit. The current perturbation was set to 5 μA.

## Data availability

The data that support the findings of this study available in this published Article and its Supplementary Information, or from the corresponding authors upon reasonable request

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

## Acknowledgements

This work was supported by the National Natural Science Foundation of China (21872016), the LiaoNing Revitalization Talents Program (XLYC1807125), the Swedish Energy Agency, and the K & A Wallenberg Foundation.

## Author contributions

F.L., T.J.M., and L.S. supervised this project; Y.Z., D.W., Q.H., and F.L. designed the experiments; Y.Z. and J.D. prepared the electrodes, performed characterization and photoelectrochemical measurements; Y.Z., D.W., F.L., and T.J.M. wrote the manuscript.

## Competing interests

The authors declare no competing interests.
