## [Peer Review File · Nature Communications]

Reviewers' Comments:

Reviewer #1:

Remarks to the Author:

I found this to be an excellent paper on the performance improvement of dye sensitized water oxidation photoanodes. The authors took the approach of replacing the ubiquitous phosphate anchoring groups with more rarely seen pyridyl anchors, with impressive results in terms of performance (particularly the high ABPE), and most importantly rationalizing this performance by using spectroscopic and electrochemical techniques to identify different mechanisms in each case with a direct consequence on performance. This control over surface reactivity is likely to be of interest to anyone working with surface-bound catalysts. From a technical point of view, the work appears solid. In my opinion, the novelty in this work lies in the notably high performance in terms of efficiency at low potential, the identification of surface dimerization as a limitation of the state-of-the-art and overcoming this limitation with an uncommonly seen alternative (the use of pyridine anchors), and as such I believe it merits publication.

One area where I do think the paper could be improved is the discussion. As noted above, the findings are significant and are well supported by the evidence, but given the journal I think more can be done to put the results in the wider context of the field. For instance, the use of pyridine anchors for surface immobilization is something I have not seen often for catalysts, so it would be good to lay out the rationale for this in terms of their use for other types of molecules (perhaps dyes) and any examples that do exist in the field of catalysis. Moreover, the apparent stronger binding of these linkers is given as the reason for their resistance to dimerization on pages 10 and 11, so references supporting this should be added, or some other rationale if none are available. Finally, the final sentence of the Discussion section on page 13 states 'Our results offer an important message for molecular level assembly design.' It would be appreciated if the authors expanded this to specify the precise message they wish readers to take away.

The authors make performance comparisons to the state-of-the-art in Table 1, demonstrating that their results for photocurrent and IPCE are indeed comparable to the best around, although not significantly exceeding them. However, these results will to a large extent be driven simply by the amount of compound present in a given area, so I am satisfied that the improvements presented in the paper (12-fold compared to catalyst '2' and 2-fold compared to '3') at near identical surface loading on the same electrodes are significant and reflect the advances made in the paper.

I also have more minor, specific suggested changes below:

Page 4 final paragraph ICP measurements. It is not discussed anywhere how ICP measurements were performed and how the results compared to the electrochemical quantification

Page 7 Oxygen quantification/Faradaic efficiency. A Faradaic efficiency (for O₂) of greater than 90% is reported and the experimental procedure adequately described. However, I feel these results could be strengthened by including the precise amount of charge passed, oxygen produced, resultant Faradaic efficiency, and how many catalytic cycles this corresponds to by dividing by the amount of catalyst measured on the surface. This would make the paper exemplary in the field, in keeping with the rest of the results presented.

Page 7 ABPE. This acronym is defined differently in the Abstract, on page 7, and in the discussion. In the abstract it is given as applied bias photon-to-current efficiency, which is the one I am most familiar with.

Page 9 paragraph 3 'The redox features for 2' closely resemble those reported previously for a surface-bound Ru-bda dimer with a 3-(pyridin-4-yloxy)propyl)phosphonic acid anchor(ref 16).' I have looked through reference 16 in the submitted manuscript and cannot find any discussion regarding dimers of Ru-bda, so I feel that this requires a little more explanation.

Page 11 middle – typo 'maxumun'

Page 12 Discussion. In addition to the improvements suggested above, there are a couple of errors. In the first paragraph, the chromophore linkages are incorrectly described as 'disphosphine' instead of 'phosphonates', and later phosphonate is misspelled ('phospnate'). At the start of the second paragraph, 'UV/vis spectrometry' should be 'UV/vis spectroscopy'.

The Methods section (along with the SI) provides a thorough description of how the experiments were performed

SI:

Methods: the units for the conductivity of MilliQ water are given as megaohms per centimeter, it should be megaohm centimeters.

The synthesis is well-described with detailed characterization, and confirms the authors' claims that the new compound is more straightforward than the phosphonate alternative.

Figure S5 – no details about how the electron microscopy were performed are given.

Reviewer #2:

Remarks to the Author:

In this manuscript, the authors presented three Ru-catalysts for water oxidation and the characterization of their loading on the TiO₂ surface. Two may undergo on-surface dimerization giving oxo-bridged dimers, diminishing the activity. The one with the 4,4'-dipyridyl anchoring ligand can maintain the monomeric structure, showing better stability with a photocurrent of 1.7 mA cm⁻². The results are very interesting and provide an important message for molecular level assembly design. The manuscript is certainly suitable for the publication in this journal, with a few clarifications/revisions added.

(1) One issue the readers may be wondering is why the three catalysts, differing only in the anchoring ligands, have very different behaviors in the dimerization. A brief explanation would be interesting.

(2) Why TiO₂|-RuP₂₊₃ with a much smaller recombination rate has a photocurrent only one-half that for TiO₂|-RuP₂₊₁? What is the potential reason?

(3) How to obtain a 2:1 ratio from Figure S10?

(4) Figure 2d: "RHE" may be "NHE".

(5) Figure 4: The words are too small to read.

(6) Figure S12: One more TiO₂ was written.

(7) Page 11: "Figure 5a" must be "Figure 5".

Reviewer #3:

Remarks to the Author:

In this communication, Zhu and co-workers describes the fabrication of a DSPEC for water oxidation with the co-adsorption on TiO₂ mesoporous film of a ruthenium tris-bipyridine bis-phosphonate as sensitizer and a ruthenium bipyridine dicarboxylate as water oxidation catalyst. DSPECs are attractive devices for water splitting, but so far their performances are much lower than those of more classical photoanodes based on medium bandgap semiconductors such as BiVO₄. In this work, the authors succeeded in achieving much higher performances than most previously reported DSPECs. They attribute this success to the fact that the catalyst is linked to the TiO₂ surface with a pyridine instead of the usual phosphonate anchoring group. The efficiency of this system is really impressive with photocurrent density of 1.7 mA/cm² at 0.1 V vs NHE giving a maximum IPCE of 25%. The higher performances were interpreted as a slower charge recombination between the injected electron in the SC and the holes on the catalyst and more importantly to the weaker tendency of the Ru catalyst to form mu-oxo-bridge dimer upon oxidation. Overall, this is an interesting discovery, but there are two points, which need to be addressed before acceptance. First, the system described in this work significantly differs from the previous DSPEC reported by the same group. Particularly, the system reported in ref. 16 of this manuscript and whose efficiency is very close was based on nano-ITO/TiO₂ core/shell oxide layers and not of nanoparticles (NPs) of TiO₂ like in this work.

The authors reported on other papers, that the nano-ITO/TiO₂ core/shell was essential to quickly remove the injected electrons from the NP and this property was crucial to achieve higher efficiency than photoanodes made of regular nanoparticles of TiO₂. It would be useful that the authors comment on this point, because this new and highly efficient system is therefore superior to the DSPEC described in the reference 16. In other words, how this new type of anchor would behave on nano-ITO/TiO₂ core/shell, in terms of photocurrent density and recombination rate constant? This is a simple experiment to realize and which could be very informative. It will unveil another features of this system, which is not raised in the discussion. Secondly, the dimerization of the Ru catalyst on the surface is significantly reduced compared to those bound via a phosphonic acid anchor. This is interpreted as a more stable binding on the TiO₂ surface with pyridine. There are several reports of TiO₂ sensitizers for DSSC bound via a pyridyl group, but few deal with the stability. However, in an article of one of the authors (Chem. Eur. J. 2012, 18, 16196), it is shown that pyridine does not seem to be more stable than carboxylic acid anchoring group. Knowing that phosphonic acid provides a much more stable linkage than carboxylic acid, it would be useful that the authors make a comparison of the relative stability between the catalysts 1, 2 and 3 on TiO₂ surface. Simple kinetic of desorption measurements in aqueous medium would shed some light on this important parameter and would support the interpretation. Without such piece of evidence, the inhibition of dimerization process with pyridyl anchor must find another interpretation.

In conclusion, this study represents a significant contribution in the field since it is a novel information that pyridine is a better anchoring group to attach such water oxidation catalyst on TiO₂. However, to influence thinking in the field and to rationally drive the development of better performing systems, the two points mentioned above need to be addressed before acceptance.

Minor points:

- 1) The letter "k" for the recombination rate constant should not be in capital letter, otherwise it creates confusion with an equilibrium constant.
- 2) On page 12, the "chelate diphosphine" should rather spell "chelate diphosphonate", idem two lines below "phosphonate derivative catalysts" and not "phosphnate derivative catalysts".

Responses to the reviewers' comments

Reviewer 1

- 1. One area where I do think the paper could be improved is the discussion. As noted above, the findings are significant and are well supported by the evidence, but given the journal I think more can be done to put the results in the wider context of the field. For instance, the use of pyridine anchors for surface immobilization is ~~something~~ I have not seen often for catalysts, so it would be good to lay out the rationale for this in terms of their use for other types of molecules (perhaps dyes) and any examples that do exist in the field of catalysis.*

Response: We appreciate the reviewer's valuable comment. The development of DSPECs were originally inspired by DSSCs, where carboxylic acid and phosphonic acid functional groups are the most commonly used anchors for surface immobilization. The acid anchors were reported to attach on TiO₂ by forming covalent bonds with the hydroxyl groups (Brønsted acid sites) on the surface. Though stable in organic media, the stability of covalent binding in aqueous solution is largely dependent on pH values due to hydrolysis in these aqueous media at pH > 5, which results in detachment of chromophore or catalyst from metal oxide.

By comparison, pyridine is known to attach on TiO₂ surfaces by coordination with the exposed Ti atoms (Lewis acid sites). Given the different binding modes between acid and pyridine sites, pyridine anchored to TiO₂ was expected to be more stable toward hydrolysis in near-neutral aqueous solutions. However, catalysis with pyridine anchors for surface immobilization is scarcely reported. Sakai and Ozawa et al. have built a TiO₂ photoanode with a pyridine-functionalized [Ru(bpy)₃]²⁺ as a chromophore (*Chem. Commun.* 2017, 53, 3042). The photoanode was examined for PEC oxidation of EDTA. Compared with the analogous phosphate-derivatized [Ru(bpy)₃]²⁺, the pyridine-functionalized [Ru(bpy)₃]²⁺ exhibited improved surface stability in aqueous solutions at pH 5. Another example was reported by the same group, where a pyridine-derivatized platinum (II) porphyrin was used as an electrocatalyst for water reduction (*Dalton Trans.* 2017, 46, 15181). In that study, the pyridine anchor was shown to be a strong linker to TiO₂ surfaces in neutral solution. Discussions related to these studies have been added to the revised manuscript to demonstrate the rationality of using pyridine anchors (Please see Page 3 for details).

2. *Moreover, the apparent stronger binding of these linkers is given as the reason for their resistance to dimerization on pages 10 and 11, so references supporting this should be added, or some other rational if none are available.*

Response: To support the strong binding capability of the pyridine linker, Ref. 20 (*Chem. Commun.* 2017, 53, 3042) and Ref. 21 (*Dalton Trans.* 2017, 46, 15181), mentioned in the response to question 1[#], have been included in the revised manuscript. In addition, the desorption kinetics of three catalysts from a TiO₂ electrode were measured in a pH 5.8 acetate buffer containing 2% sodium ascorbate. Our results revealed a dynamic TiO₂|catalyst interface, agreeing with the findings obtained by Sakai's group (Please see the discussion on Page 12 of the text and Figure S14 and the experimental details in the supporting information). The formation of surface-bound "green dimer" is proposed to be a result of desorption of oxidized phosphonate-modified catalysts, followed by dimerization and re-adsorption within mesoporous TiO₂.

In agreement with this hypothesis, dimerization of immobilized catalyst has been reported by Reisner et. al. (*Angew. Chem. Int. Ed.* 2016, 55, 7388). In their work, a surface-bound Mn-Mn dimer as catalytic active species for CO₂ reduction was formed by the temporary desorption of a phosphonate-modified monomeric Mn catalyst, followed by dimerization and re-anchoring on TiO₂. A brief discussion has been added in Page 12.

3. *Finally, the final sentence of the Discussion section on page 13 states 'Our results offer an important message for molecular level assembly design.' It would be appreciated if the authors expanded this to specify the precise message they wish readers to take away.*

Response: Thank you for this kind suggestion. The sentence “Recognizing the importance of anchoring groups in determining the catalytic active species on metal oxide surface adds a new approach to the rational tuning of hybrid interfaces for DSPEC applications.” has been added to the conclusions.

4. *The authors make performance comparisons to the state-of-the-art in Table 1, demonstrating that their results for photocurrent and IPCE are indeed comparable to the best around, although not significantly exceeding them. However, these results will to a large extent be driven simply by the amount of compound present in a given area, so I am satisfied that the improvements presented in the paper (12-fold compared to catalyst '2' and 2-fold compared to '3') at near identical surface loading on the same electrodes are significant and reflect the advances made in the paper.*

Response: Thanks for the comment. Unfortunately, the exact loading of catalysts and dyes on TiO₂ were not provided by the authors in the reference papers listed in Table 1. In this study, however, catalysts **1**, **2** and **3** were loaded on TiO₂-**RuP** with similar surface coverage around 1.2×10^{-8} mol cm⁻² to make a reasonable comparison of their PEC performances.

5. *Page 4 final paragraph ICP measurements. It is not discussed anywhere how ICP measurements were performed and how the results compared to the electrochemical quantification.*

Response: For the electrode TiO₂-**RuP**²⁺, the amount of **RuP**²⁺ on TiO₂-**RuP**²⁺ was measured by ICP before catalyst loading. A detailed procedure for ICP measurements is provided in the supporting information. First, **RuP**²⁺ adsorbed on TiO₂ was desorbed by 1M NaOH and the eluent was diluted with H₂O into a 5 mL solution. The atomic emission of Ru in solution was then recorded by ICP followed by quantification based on a standard curve of atomic emission intensity as a function of Ru concentration. With this method, the loading of **RuP**²⁺ was estimated as 6×10^{-8} mol cm⁻². The surface coverage of **RuP**²⁺ on TiO₂ was also evaluated by electrochemical quantification based on the area of Ru^{III/II} redox couple (please see Figure S6b in the supporting information). With this method, a similar coverage of 5.92×10^{-8} mol cm⁻² for **RuP**²⁺ was measured. Therefore, both methods are reliable for analyzing the surface coverage. However, the ICP approach is not applicable to

the catalyst added on the electrode following RuP loading, due to the disturbance of ruthenium cations from the chromophore. Therefore, the evaluation of catalyst coverage relied on the electrochemical method.

6. *Page 7 Oxygen quantification/Faradaic efficiency. A Faradaic efficiency (for O₂) of greater than 90% is reported and the experimental procedure adequately described. However, I feel these results could be strengthened by including the precise amount of charge passed, oxygen produced, resultant Faradaic efficiency, and how many catalytic cycles this corresponds to by dividing by the amount of catalyst measured on the surface. This would make the paper exemplary in the field, in keeping with the rest of the results presented.*

Response: Thank you for your kind suggestion. The precise amount of charge passed, oxygen produced, Faradaic efficiency and catalytic cycles have been added to the first paragraph of Page 8 as “During this process, 6.24 μ mol evolved oxygen was quantified by gas chromatography with 2.66 C charges passed through the photoanode, corresponding to a faradaic efficiency of over 90% and a TON of 520”.

7. *Page 7 ABPE. This acronym is defined differently in the Abstract, on page 7, and in the discussion. In the abstract it is given as applied bias photon-to-current efficiency, which is the one I am most familiar with.*

Response: We are sorry for the inconsistency. ABPE in this paper represents the applied bias photon-to-current efficiency, which was wrongly expressed on Page 7.

8. *Page 9 paragraph 3 'The redox features for 2' closely resemble those reported previously for a surface-bound Ru-bda dimer with a 3-(pyridin-4-yloxy)propyl)phosphonic acid anchor(ref 16).' I have looked through reference 16 in the submitted manuscript and cannot find any discussion regarding dimers of Ru-bda, so I feel that this requires a little more explanation.*

Response: We are deeply sorry for making a mistake on the Ref. number, which should be [44] (*J. Am. Chem. Soc.* 2019, 141, 7926 -7933) instead of [16]. In Ref. 44, the Ru-bda catalyst decorated with 3-(pyridin-4-yloxy)propyl)phosphonic acid anchor was supported on a core-shell TiO₂ film, air oxidation of the supported catalyst exhibited a characteristic absorption at 690 nm, which was ascribed to the μ -oxo-bridged Ru-O-Ru dimer. In addition, a similar Ru-O-Ru dimer formed by electrochemical oxidation of the monomeric Ru-bda complex has been reported by Concepcion and co-workers in *Chem. Commun.* 2015, 51, 4105-4108 (Ref. 43). These results suggest that the formation of dimer is an intrinsic feature for phosphonate-decorated Ru-bda catalysts under oxidative conditions.

9. Page 11 middle – typo ‘maxumun’

Response: The typo has been corrected in the revised manuscript (Page 12 in the revised version).

10. Page 12 Discussion. In addition to the improvements suggested above, there are a couple of errors. In the first paragraph, the chromophore linkages are incorrectly described as ‘disphosphine’ instead of ‘phosphonates’, and later phosphonate is misspelled (‘phospnate’). At the start of the second paragraph, ‘UV/vis spectrometry’ should be ‘UV/vis spectroscopy’.

Response: Thanks for pointing out these errors, they have been corrected in the revised manuscript.

11. The Methods section (along with the SI) provides a thorough description of how the experiments were performed. Methods: the units for the conductivity of MilliQ water are given as megaohms per centimeter; it should be megaohm centimeters.

Response: Thanks for pointing out this error and it has been corrected.

12. Figure S5 – no details about how the electron microscopy were performed are given.

Response: The details for the electron microscopy measurements have been provided in the supporting information (Please see Page 2, paragraph 2).

Reviewer 2

1. One issue the readers may be wondering is why the three catalysts, differing only in the anchoring ligands, have very different behaviors in the dimerization. A brief explanation would be interesting.

Response: The different behaviors of catalysts were proposed to arise from their lability on TiO₂. To evaluate the stability of **1**, **2** and **3** on electrodes, a desorption test of the TiO₂|catalyst was carried out in a pH 5.8 acetate buffer containing 2% sodium ascorbate. Experimental results show that catalysts **2** and **3** decorated with the phosphonate anchor are prone to detachment from the electrode surface with a desorption rate constant k_{des} of 270 s⁻¹ for **2** and 63 s⁻¹ for **3**, while catalyst **1** decorated with the pyridine anchor is much more stable on the surface ($k_{des}(\mathbf{1})=1.4$ s⁻¹) (Please see the discussion on Page 12 of the text and Figure S14 and the experimental details in the supporting information). These results demonstrate a dynamic TiO₂|catalyst interface, the formation of surface-bound “green dimer” is

assumed to be a result of desorption of oxidized phosphonate-modified catalysts under the conditions of PEC experiments, followed by dimerization and re-adsorption within mesoporous TiO₂.

In support of this hypothesis, the dimerization of immobilized catalyst has been reported by Reisner et al. (*Angew. Chem. Int. Ed.* 2016, 55, 7388). In their work, a surface-bound Mn-Mn dimer as the catalytically active species for CO₂ reduction was formed by temporary desorption of a phosphonate-modified monomeric Mn catalyst, followed by dimerization and re-anchoring on TiO₂. A brief explanation has been added in the last paragraph of Page 12.

2. *Why TiO₂-RuP²⁺,3 with a much smaller recombination rate has a photocurrent only one-half that for TiO₂-RuP²⁺,1? What is the potential reason?*

Response: Back electron transfer (recombination) from the conduction band of TiO₂ to the oxidized chromophore or catalyst is a major limitation to the efficiency of DSPECs, however, other factors also play important roles on PEC performance. The photocurrent is determined by the overall results of hole transfer (catalysis) and charge recombination. Though catalyst **3** has a small recombination rate, by taking advantage of long alkyl chains against back electron transfer, the surface-bound active species also poses a significant effect on the photocurrent. Spectroelectrochemical experiments revealed the dimerization of **3** on TiO₂ surfaces (Figure 5) during the PEC reaction while **1** is stable against dimerization. Based on the CV in Figure S13, the in-situ generated dimer **3'** exhibited a lower catalytic activity than its monomeric analogue, and hence a lower photocurrent.

3. *How to obtain a 2:1 ratio from Figure S10?*

Response: The evolved gas was quantified by GC by using an external standard method, a detailed procedure has been provided in the “Gas Chromatography Measurements” section of the supporting information. In Figure S10, the integrated peak areas for oxygen and hydrogen are 1203.2 and 24382.8, respectively, corresponding to molar amounts of 12.7 μmol for hydrogen and 6.24 μmol for oxygen using the calibration coefficient obtained from the standard curve for each gas which gave a 2:1 ratio of hydrogen to oxygen.

4. *Figure 2d: “RHE” may be “NHE”.*

Response: The horizontal axis in Figure 2d is indeed represented by RHE, not NHE, which is converted from the potential measured with Ag/AgCl as the reference electrode *via* the equation $V_{\text{RHE}} = V_{\text{Ag/AgCl}} + 0.059 \text{ pH} + 0.197$. The value of ABPE

was calculated based on a standard equation that is widely used for PEC cells,

$$ABPE(\%) = \frac{(J_{light} - J_{dark}) \times (1.23 - V_{RHE})}{P_{light}} \times 100\%$$

Where 1.23 is the theoretical potential of water splitting based on RHE. The use of RHE enables us to conveniently compare the efficiencies of PECs operating at various pHs.

5. *Figure 4: The words are too small to read.*

Response: The Font size in Fig. 4 has been adjusted.

6. *Figure S12: One more TiO₂ was written.*

Response: Sorry for our carelessness and it has been corrected.

7. *Page 11: "Figure 5a" must be "Figure 5".*

Response: Thanks for pointing out this mistake and it has been corrected in the revised manuscript.

Reviewer 3

1. *First, the system described in this work significantly differs from the previous DSPEC reported by the same group. Particularly, the system reported in ref. 16 of this manuscript and whose efficiency is very close was based on nano-ITO/TiO₂ core/shell oxide layers and not of nanoparticles (NPs) of TiO₂ like in this work. The authors reported on other papers, that the nano-ITO/TiO₂ core/shell was essential to quickly remove the injected electrons from the NP and this property was crucial to achieve higher efficiency than photoanodes made of regular nanoparticles of TiO₂. It would be useful that the authors comment on this point, because this new and highly efficient system is therefore superior to the DSPEC described in the reference 16. In other words, how this new type of anchor would behave on nano-ITO/TiO₂ core/shell, in terms of photocurrent density and recombination rate constant? This is a simple experiment to realize and which could be very informative. It will unveil another feature of this system, which is not raised in the discussion.*

Response: Thanks for the valuable suggestion. We have attempted to immobilize the catalyst with pyridine anchor (**1**) on core/shell ITO/TiO₂ thin films. Unfortunately, the maximum loading amount of **1** on ITO/TiO₂ was found to be only one-sixth of that on TiO₂ film (Figure R1 and Figure S6a). The low catalyst loading led to a photocurrent of 0.35 mA/cm² for ITO/TiO₂|-**RuP**, **1** at 0.2 V vs NHE (Figure

R2), far lower than that for TiO₂|**-RuP, 1** (1.7 mA/cm² under the same conditions).

Figure R1. Analysis of peak current of ITO/TiO₂|**-RuP, 1**, in 0.1 M acetic acid/acetate buffer at pH 5.8 containing 0.5 M NaClO₄ at a scan rate 10 mV/s.

Figure R2. Current density-time traces for core/shell ITO/TiO₂|**-RuP²⁺, 1** at a constant bias of 0.2 V vs. NHE with illumination at > 400 nm at 100 mW cm⁻² at pH 5.8 in a 0.1 M acetate buffer containing 0.5 M NaClO₄.

A plausible reason behind this effect is the different binding modes adopted by pyridine and acid anchors. Pyridine anchoring groups are attached on TiO₂ by a preferential coordination with the exposed Ti sites (Lewis acid sites), while phosphonate anchoring group is attached by covalent bonding with the hydroxyl groups on TiO₂ surface (Brønsted acid sites). The nano-ITO/TiO₂ core/shell film, prepared by atomic layer deposition (ALD), prefers to form anatase-phase TiO₂. The ordered and regular anatase TiO₂ crystal with fewer surface defects can benefit charge separation, but leaves fewer unsaturated coordinated Ti atoms on the surface compared to the amorphous TiO₂ used in this paper. As a result, core/shell ITO/TiO₂ samples show limited abilities for pyridyl binding.

2. Secondly, the dimerization of the Ru catalyst on the surface is significantly reduced compared to those bound via a phosphonic acid anchor. This is interpreted as a more stable binding on the TiO₂ surface with pyridine. There are several reports of TiO₂ sensitizers for DSSC bound via a pyridyl group, but few deal with the stability. However, in an article of one of the authors (*Chem. Eur. J.* 2012, 18, 16196), it is shown that pyridine does not seem to be more stable than carboxylic acid anchoring group. Knowing that phosphonic acid provides a much more stable linkage than carboxylic acid, it would be useful that the authors make a comparison of the relative stability between the catalysts 1, 2 and 3 on TiO₂ surface. Simple kinetic of desorption measurements in aqueous medium would shed some light on this important parameter and would support the interpretation.

Response: Thank you for your kind suggestions. We noticed the article published in *Chem. Eur. J.* 2012, 18, 16196. In that paper, the stability of dyes with various anchors were evaluated in anhydrous acetonitrile. In our case, the DSPEC photoanode was immersed in aqueous acetate buffer. The solvent-dependent stability of anchoring groups in organic and aqueous media has been reported previously (Meyer et al. *ACS Appl. Mater. Interfaces* 2012, 4, 146). Though carboxylic acid or phosphonic acid anchors are known to form relatively stable covalent bonds with hydroxyl groups on TiO₂ in organic media, they tend to detach from metal oxides due to hydrolysis in aqueous media. In a recent paper, the adsorption stabilities of [Ru(bpy)₃]²⁺ derivatives with phosphoric acid, carboxylic acid and pyridine anchoring groups in a pH 5 aqueous solution have been compared (*Chem. Commun.* 2017, 53, 3042). The results showed almost quantitative desorption of the [Ru(bpy)₃]²⁺ complex with carboxylic acid anchor within 10 min and the desorption of phosphate derivatized [Ru(bpy)₃]²⁺ complex by 30% in 2 h. By contrast, the desorption of pyridine derivatized [Ru(bpy)₃]²⁺ was negligible for at least for 2 h. The same trend was also observed in neutral (pH 7.0) or basic (pH 9.0) buffer solutions, indicating pyridine anchors suitable for use in DSPEC cells in aqueous solution.

In response to the suggestions of this reviewer and reviewer 2[#], the desorption kinetics of three catalysts from TiO₂ electrodes were measured in a pH 5.8 acetate buffer with 2% sodium ascorbate (Please see the discussion on Page 12 of the text and Figure S14 and the experimental details in the supporting information). Our experimental results showed that catalysts 2 and 3, with phosphonate anchors are prone to detachment from the electrode surface with a desorption rate constant k_{des} of 270 s⁻¹ for 2 and 63 s⁻¹ for 3, while catalyst 1 decorated with the pyridine anchor is much more stable on the surface (k_{des} (1) = 1.4 s⁻¹). Under the conditions for the

PEC experiments, the formation of surface-bound “green dimer” is proposed to be a result of desorption of oxidized phosphonate-modified catalysts, followed by dimerization and re-adsorption within mesoporous TiO₂.

In support of this hypothesis, the dynamic TiO₂|catalyst interface and dimerization of immobilized catalyst has been reported by Reisner et. al. (*Angew. Chem. Int. Ed.* 2016, 55, 7388). In their work, a surface-bound Mn-Mn dimer, as a catalytically active species for CO₂ reduction, was formed by the temporary desorption of a phosphonate-modified monomeric Mn catalyst, followed by dimerization and re-anchoring on TiO₂. These results and the related discussion have been added to the revised manuscript (Page 12).

3. *The letter “k” for the recombination rate constant should not be in capital letter, otherwise it creates confusion with an equilibrium constant.*

Response: It has been corrected according to the reviewer’s suggestion.

4. *On page 12, the “chelate diphosphine ” should rather spell “chelate diphosphonate”; idem two lines below “phosphonate derivative catalysts ” and not “phospnate derivative catalysts ”.*

Response: These errors have been corrected in the revised manuscript.

Reviewers' Comments:

Reviewer #1:

Remarks to the Author:

The authors have done a thorough job in responding to my comments - thank you. I think the quality of the discussion now matches that of the excellent experimental work presented. I happily recommend publication.

Reviewer #2:

Remarks to the Author:

The authors have addressed the issues raised. I now recommend the acceptance of the revised manuscript.

Reviewer #3:

Remarks to the Author:

First, I thank the authors for the taking care of the remarks and addressing them satisfyingly. My concerns, namely about the superiority of the binding stability with pyridine anchor, have been cleared away and rationalized. This study now really deserves to be published as it is in Nature Communication owing to the significant impact of the results presented.